# Preparing future STEM faculty through flexible teaching professional development

Bennett B. Goldberg[1]☯*, Derek O. Bruff[2]☯¤a, Robin McC. Greenler[3], Katherine Barnicle[3], Noah H. Green[4]¤b, Lauren E. P. Campbell[5], Sandra L. Laursen[6], Matthew J. Ford[7]¤c, Amy Serafini[8], Claude Mack[5]¤d, Tamara L. Carley[9], Christina Maimone[7], Henry (Rique) Campa, III[10]

1 Department of Physics and Astronomy, Northwestern University, Evanston, IL, United States of America, 2 Center for Teaching, Vanderbilt University, Nashville, TN, United States of America, 3 CIRTL Network, University of Wisconsin–Madison, Madison, WI, United States of America, 4 Green Scientific and Educational Consulting, Charlottesville VA, United States of America, 5 Department of Physics & Astronomy, Vanderbilt University, Nashville, TN, United States of America, 6 Ethnography & Evaluation Research, University of Colorado Boulder, Boulder, CO, United States of America, 7 Northwestern IT Research Computing Services, Northwestern University, Evanston, IL, United States of America, 8 Department of Educational Foundations, Leadership & Technology, Auburn University, Auburn, Alabama, United States of America, 9 Department of Geology and Environmental Geosciences, Lafayette College, Easton, PA, United States of America, 10 Graduate School and Department of Fisheries and Wildlife, Michigan State University, East Lansing, MI, United States of America

☯ These authors contributed equally to this work.
¤a Current address: Center for Excellence in Teaching and Learning, University of Mississippi, Oxford, MS, United States of America
¤b Current address: The Science Communication Lab, Berkeley, CA, United States of America
¤c Current address: School of Engineering & Technology, University of Washington Tacoma, Tacoma, WA, United States of America
¤d Current address: Machtfit GmbH, Berlin, Germany
* bennett.goldberg@northwestern.edu

**Data Availability Statement:** Data and code are available at https://doi.org/10.5281/zenodo.7508428

## Abstract

We have prepared thousands of future STEM faculty around the world to adopt evidence-based instructional practices through their participation in two massive open online courses (MOOCs) and facilitated in-person learning communities. Our novel combination of asynchronous online and coordinated, structured face-to-face learning community experiences provides flexible options for STEM graduate students and postdoctoral fellows to pursue teaching professional development. A total of 14,977 participants enrolled in seven offerings of the introductory course held 2014–2018, with 1,725 participants (11.5% of enrolled) completing the course. Our results of high levels of engagement and learning suggest that leveraging the affordances of educational technologies and the geographically clustered nature of this learner demographic in combination with online flexible learning could be a sustainable model for large scale professional development in higher education. The preparation of future STEM faculty makes an important difference in establishing high-quality instruction that meets the diverse needs of all undergraduate students, and the initiative described here can serve as a model for increasing access to such preparation.

**Funding:** This work was supported by the National Science Foundation under grant number DGE 1347605.

**Competing interests:** The authors have declared that no competing interests exist.

## Introduction

There is recognition that evidence-based, student-centered instruction in science, technology, engineering, and mathematics (STEM) generally increases undergraduate student learning and success in STEM [1, 2] and reduces the disparities in outcomes between marginalized students who are historically underrepresented in STEM and majority students in STEM [3–6]. There is also evidence that current [7, 8] and future [9] faculty who engage in effective teaching professional development go on to implement evidence-based pedagogies in their classes. These findings motivate pedagogical professional development programs offered by university teaching centers, graduate schools, and postdoctoral training initiatives [10].

Future STEM faculty—that is, doctoral students and postdoctoral fellows, hereafter referred to as postdocs, who seek academic careers—face particular challenges in learning about and adopting evidence-based teaching practices, including limited opportunities and lack of advisor support for pedagogical professional development [11–14]. Despite this, graduate students and postdocs may be more receptive than current faculty to explore and implement evidence-based teaching practices because they are in the process of learning the standards of academia, developing scientific and teaching practices in their discipline, and are preparing for competitive academic positions [15]. Encouragingly, future STEM faculty who participate in moderate- or high-engagement pedagogical professional development (greater than 25 hours of participation) report significantly improved self-efficacy as instructors and significantly higher adoption of evidence-based teaching practices [9] and perform as well or better in research [16].

To provide such professional development opportunities to future STEM faculty, and thereby improve undergraduate education in the U.S. more broadly, the Center for the Integration of Research, Teaching, and Learning (CIRTL) Network, which currently consists of 43 research universities across the United States and Canada, provides structured pedagogical professional development programs for graduate students and postdocs at individual campuses and through cross-Network programming [17–19]. Many of these programs are structured as in-person or virtual, synchronous learning communities [18, 20], where participants meet to learn from and with each other as they pursue shared learning goals [21]. The Network also serves as a community of practice [22] for leaders of future STEM faculty development to share strategies and expertise and co-develop and implement network-wide programs.

In 2013, a series of CIRTL Network conversations on emerging models for future STEM faculty development led a small group of faculty, administrators, and researchers to propose a new initiative centered on using massive open online courses, also known as MOOCs. Interest in this new form of asynchronous online education accelerated rapidly in the early 2010s [23], with educators and researchers exploring the potential for online tools such as videos, discussions, and peer assessments to support learning for thousands of concurrent students [24]. Interestingly, research shows that the more successful MOOCs have been associated with targeted rather than general audiences [25].

In this context of pedagogical experimentation and with funding from the National Science Foundation, our specific goals were to design, deliver, and evaluate the use of MOOCs on evidence-based undergraduate STEM teaching for future faculty pedagogical professional development. This in itself was not novel; other MOOCs developed in the same time frame also had this focus [26]. Inspired by instructors who "wrapped" campus-based courses around existing MOOCs [27] and informed by the CIRTL Network's experience with campus-based and virtual learning communities, we planned the online courses to be delivered in three different modes to meet the diverse learning needs of future faculty: (1) as stand-alone MOOCs for online participants, (2) as blended online and in-person experiences constructed with what we called MOOC-Centered Learning Communities, or MCLCs, and (3) as open educational

resources for use by individuals or by campus-based professional development programs. By inviting colleagues around the CIRTL Network and beyond to host MCLCs of participants in the online courses and providing MCLC facilitators with learning guides to support their local in-person meetings, we designed a novel structure that has enabled us to meet the professional development needs of thousands of future STEM faculty worldwide.

To inform the development of our voluntary educational program for adult learners, we gathered quantitative and qualitative data to evaluate our program, to understand our audience, and to discover the outcomes they derived (or not) from participating. Like most questions about "what works" in education, the answers necessarily depend on who participates and how [28]. Our data and analyses offer relevance beyond our own program, however, because they shed light on fundamental questions—still very much debated by researchers—about the affordances, limitations and challenges for MOOCs in facilitating education for learners in diverse, global contexts [29–31]. Indeed, a recent review suggests that pedagogical approach and educational resources of MOOCs—aspects we emphasize are particularly under-studied [32]. Hence our purpose is to contribute to this body of knowledge a careful description of a particular educational design and a critique of whether and how the resulting MOOC and associated MCLCs served our intended audience. We operationalized this purpose with three guiding questions:

1. Will our model enroll participants of our intended audience (future STEM faculty) at a scale beyond that typically reached by traditional on-campus or synchronous professional development programs?

2. To what extent will participants increase their knowledge of and confidence in delivering effective STEM instruction?

3. What effects will the MCLCs have on participation and participant outcomes in the broader MOOC?

These questions recognize the exploratory nature and context sensitivity of studies about real-world educational practices, in contrast to studies involving hypothesis testing in structured environments where variables can be controlled and individually manipulated [33].

## Methods

The methods section is organized as follows: We first describe the course structure, curriculum design, assessments, and associated MCLCs of the two courses, an introductory and advanced MOOC, in order to situate the guiding questions of participant engagement, participant learning, and MCLC outcomes within the project design. We next describe the data sources of participant course activity, pre- and post-course surveys, and MCLC facilitator surveys and focus groups that we used to answer our guiding questions and inform our conclusions.

### Course structure and logistics

In 2013–2014, we launched an eight-week introductory MOOC, *Introduction to Evidence-based Undergraduate STEM Teaching*, followed by a second, more advanced eight-week MOOC, *Advanced Learning Through Evidence-Based STEM Teaching*, in 2015–2016. Each course consists of six modules, each featuring instructional videos, discussion prompts, recommended readings, and a quiz that together take 3-to-5-hours to complete. Each course also included three peer-graded assessments (PGAs). Each course took approximately a year to design and build, and each course drew on expertise from multiple institutions and individuals within and outside the CIRTL Network.

The introductory course examines the fundamentals of learning and learning design, including learning objectives, assessment, and active learning, culminating with a final PGA in which participants develop a sample lesson plan incorporating these core elements. The advanced course delves deeper into evidence-based teaching practices, including peer instruction, cooperative learning, and inquiry-based labs. The final PGA in the second course requires participants to develop a teaching philosophy statement that demonstrates their understanding of and preferences among the teaching practices they have learned. Based on course objectives, module learning goals and assessments, we defined a "completer" as a participant who completed a combination of quizzes (weighted 60% together for four highest scores) and PGAs (individually weighted 10%, 10%, and 20%) with an overall score of at least 50%. Each course has been offered once or twice a year since they launched, originally on the Coursera platform and now on the edX platform.

To foster greater engagement and learning, we encourage participants in the online courses to join or start an MCLC. These learning communities, typically hosted on university campuses, meet weekly to share, discuss, and contextualize what participants are learning in the online course. Depending on local needs, MCLCs can be part of credit-bearing courses or non-credit seminars or simply be an informal set of meetings among peers or colleagues. Each MCLC has a facilitator who regularly convenes the community and plans discussions or other activities for the in-person meetings [34]. We provide an "MCLC Facilitators' Guide" to support MCLC facilitators that includes learning goals and objectives for online and in-person sessions; overviews of online videos, discussion prompts to engage participants with course content, and assignments; and 3–7 suggested activities with facilitator notes for each module that complement and extend the online materials. Our project team markets the potential of MCLCs for professional development associated with teaching and learning and recruits MCLC facilitators at CIRTL Network campuses and through our respective networks (e.g., professional disciplinary societies, campus academic units, colleagues external to our own campuses) to draw a diverse and international community.

The project website, https://www.stemteachingcourse.org/, makes freely available most of the course content, including videos, accompanying slides, discussion prompts, and instructions for each PGA. The project also has a public YouTube channel, https://www.youtube.com/user/cirtlmooc, featuring all course videos organized by course module. All materials are made available under a Creative Commons 4.0 Attribution-Noncommercial license to facilitate reuse by anyone interested in STEM teaching or pedagogical professional development.

## Data sources and analysis

We invited course participants to take surveys for project evaluation. These surveys and learning activity data obtained from the course platforms were determined "not human subjects research" by the Michigan State University Institutional Review Board (IRB), hence formal consent was not required. Interviews of MCLC facilitators were determined to be Exempt by the IRB at University of Colorado Boulder (protocol 15–0658). All interview respondents provided informed consent in writing.

Data on course participation, engagement, and outcomes comes from four main sources, the analysis of which are used to answer our first two guiding questions on whether and how we reached our target audience and what participants learned. The first is the MOOC platforms, Coursera and edX. For both platforms, data are available on which quizzes and PGAs participants attempted and completed, which course videos they watched (operationalized as initiation of playing), and whether they officially completed the course. The second source of data is a pre-course survey, which asked about participants' demographic characteristics, intended course activity, and familiarity with concepts covered in the course.

The third source of data is a post-course survey, which inquired about demographics, self-reported activity in the course, familiarity with concepts covered in the course, self-reported learning gains, and evaluations of course components such as assignments and videos. The pre- and post-course surveys were first administered through Coursera and then later Qualtrics (Qualtrics, Provo, UT). Course participants were encouraged, but not required, to take the surveys. Respondents were asked to generate and enter an anonymous code to allow linkage between the two surveys. Data were extracted from the course platforms and Qualtrics, cleaned, and combined in a database and then analyzed by a combination of descriptive and statistical analysis. Descriptive analysis was conducted with Python, including with the pandas library [35], and data visualizations were created with seaborn [36] and matplotlib [37]. To answer our first Guiding Question of whether we reached our target audience and had an impact at a larger scale than local projects, we analyzed the demographic and completion data, and compared to local professional development. To answer our second Guiding Question of the learning participants achieved, we examined the course assessments and self-reported gains.

Finally, data about MCLC facilitators came from surveys of facilitators conducted in 2014–15 and interviews conducted in 2016, and are used to answer our third Guiding Question Analysis of these provided rich formative feedback and also characterized MCLC implementation. The surveys gathered information about MCLC composition, structure, and participation, and about the facilitators' preparation, interactions with learners, and benefits they perceived for the learners and to themselves [38]. Interviews probed these elements further and asked facilitators about the MOOC elements, MCLC facilitation strategies, and integration with other professional development programming at their campus [39].

## Results

### Learner engagement and target audience

To answer our first Guiding Question we sought to understand whether we reached our intended audience, how they engaged, what they learned, what motivated them, and their overall satisfaction. A total of 14,977 people enrolled in seven offerings of the introductory course held 2014–2018, with 1,725 participants from approximately 60 countries completing. (As noted above, course participants who completed a combination of quizzes and PGAs earned a certificate of completion.) The average introductory course completion rate was 11.5% overall. Enrollment and completion numbers for the advanced MOOC were lower. Overall, 5,320 total people registered for the four offerings to date of the advanced MOOC, with 291 completers and a course-averaged 6.3% completion rate (S1 Table in S1 File).

Pre-course survey data from a subset of registrants (described in more detail below) offer insight into course participants' goals in taking the course. Respondents' most frequently reported intent was to "enhance my STEM teaching skills," with 93% ranking this as "Important" or "Very important," with the goal to "enhance learning of my students" close behind at 89%. These and other factors related to improved teaching and learning substantially outranked achievement-oriented goals including "earn credentials for my CV" (49%) and "earn a statement of accomplishment," (31%). The importance of motivating factors did not differ notably by role or by the respondent's ultimate level of engagement with the course. S6 Table in in S1 File shows the rank order of all motivations.

In addition to the category of "completer," which we defined a priori, platform data pointed us to additional categories of engagement and learning, which we define from the perspective of MOOCs and similar free, online, self-directed adult learning environments. In developing a "learner" category, we focused on those who engaged in significant ways with the course

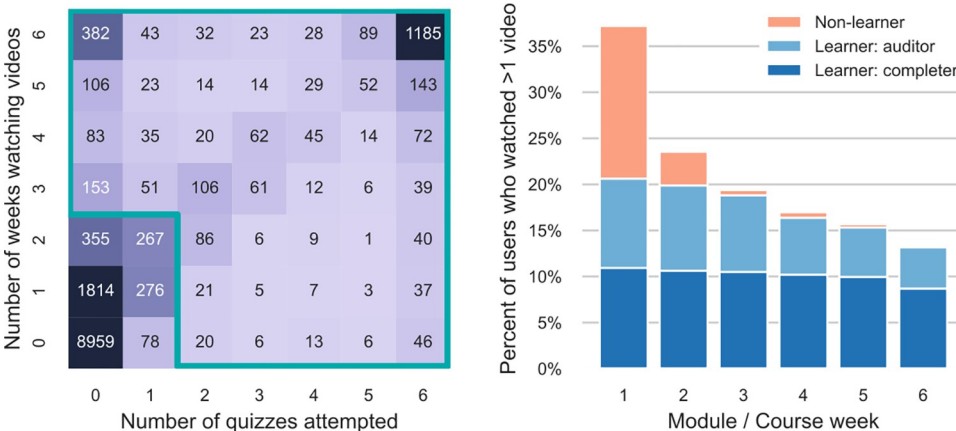

**Fig 1. Learner engagement v. video watching, assignments, and weeks of the introductory STEM MOOC.** (A) Joint histogram of enrolled in the introductory MOOC by total number of modules/course weeks in which they participated by watching videos (vertical axis) or taking quizzes (horizontal axis). The outlined region approximately separates "learners" from disengaged "non-learners;" a small number (31 or 1%) of learners who completed a PGA but few quizzes may not fall within the outlined region. (Note that the course included a module "0", introducing the course.) (B) Percent of total enrolled who participated during each module/course week by watching more than one video that week, distinguished by character of participant.

materials, as a proxy for learning and gains in knowledge. We set the threshold for "learners" based on greater than 50% drop-offs observed in participant activity week-to-week as a function of both video watching and assignment completion: "learners" watched course videos after Week 2 and/or completed course quizzes after Week 1 (the outlined region in Fig 1A). As other MOOC developers have also found, course engagement typically drops off after the first or second week [40, 41]. Learners represented 22% of those enrolled in the introductory course and were distinguished by their behaviors into two main groups: completers (53% of learners, or 11.5% of all enrolled) who participate in quizzes and peer-graded assignments, and non-completers, whom we call "auditors" (our term, representing 47% of learners, 10.2% of enrolled). Auditors primarily watch course videos without attempting quizzes and PGAs, and about half the auditors watched videos in all six of the course modules [42–44]. Thus, 68% of the learners engaged continuously with the material throughout the course. Fig 2 characterizes

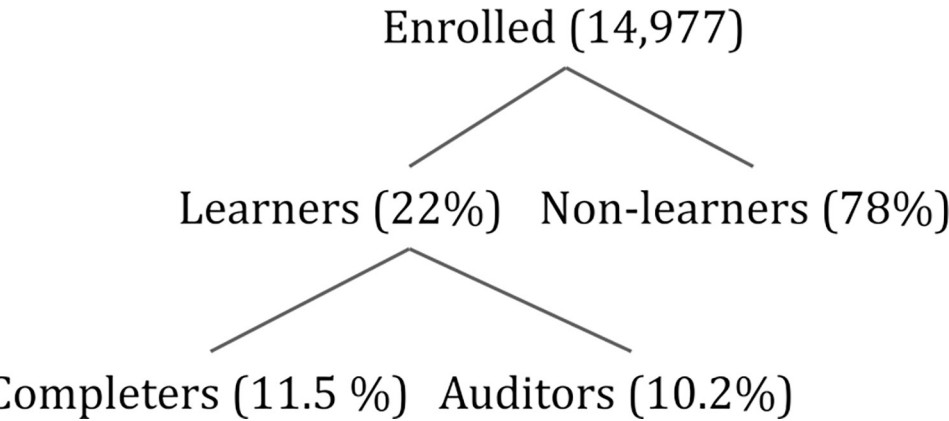

**Fig 2. Consort diagram of course participants.** Consort diagram displaying the overall total of participants who enrolled, the fraction that engaged or didn't in more than one video, page or activity, and the subsequent fraction that we describe as learners, both those who completed and those who audited.

these engagements in a consort diagram with the percentages of the introductory course. The supporting materials contain detailed data about week-by-week engagement with course activities, and additional data that supports the distinction between learners and non-learners, completers and auditors.

Pre- and post-course respondents were largely PhDs and postdocs (50% pre-course and 58% post-course) with faculty an additional 20% pre-course and 16% post-course, with the remaining other category (27% pre-course and 19% post-course) a mix of staff and non-academics Nearly all (91%) of the pre-course survey respondents indicated their disciplines from STEM or Social, Behavioral, and Economic Sciences (SBES) fields. A majority identified as female (60% pre-course/56% post-course), and nearly half came from one of the 38 CIRTL universities. Nearly one third of post-survey respondents participated in an MCLC. We are, therefore, confident that the course is reaching future STEM faculty or professionals, that the connection to the CIRTL Network has been instrumental in disseminating the training professional development program, and that the course also reaches many learners beyond the CIRTL Network (see S1 and S2 Tables in S1 File).

Across seven instances of the introductory course, 3,884 students (26% of enrolled) took the pre-course survey. In a subset of the data where we can link survey participation to course engagement behaviors, pre-course survey respondents included 57% of learners in the course; conversely, 55% of pre-course survey respondents engaged with the course as learners. Similarly, at the end of the course, half (55%) of the completers responded to the post-course survey; among post-survey respondents, 84% completed the course and an additional 8% engaged during all six modules. Results from the surveys are, therefore, very reflective of the experiences and demographics of learners and course completers (see S7 Fig in S1 File for full details).

Overall, 34% of pre-course survey respondents completed the course. Among them, postdoctoral researchers completed at a higher rate of 39% compared to 32% for other participants (Fisher's exact test, $p = 0.005$). Those who indicated on the pre-course survey that they intended to pursue an academic career (74% of respondents) completed at a significantly higher rate, 37%, than those who did not, 23% (Fisher's exact test, $p \ll 0.001$).

## Learning and confidence to apply content

To address Guiding Question 2, we asked course participants to rate their learning gains in seven areas that reflected our intended general outcomes for them. On average, respondents to the post-course survey rated four gains in the range of "good" to "great" gains: Confidence to implement teaching and learning strategies covered in class; Interest in additional classes related to teaching and learning; Interest in discussing teaching and learning with colleagues and friends; and Confidence that they understand the material covered (S9 Fig in S1 File). The remaining gains were rated, on average, "moderate" to "good": Enthusiasm for STEM teaching and learning; Interest in an additional MOOC related to teaching and learning; and willingness to seek help from faculty or peers regarding teaching and learning. No gains were rated lower than moderate. Additionally, participants who responded to both pre- and post-course surveys reported higher post-course familiarity with several specific concepts taught in the course, including backwards design, setting learning objectives, use of formative and summative assessments, and leveraging diversity to enhance teaching and learning. Fig 3A displays these results as the average change and Fig 3B as paired. Supporting materials include additional analysis of which course elements respondents found helpful to their learning, self-reported gains in interest in course topics, confidence in applying skills covered in the course, and post-course familiarity with key concepts.

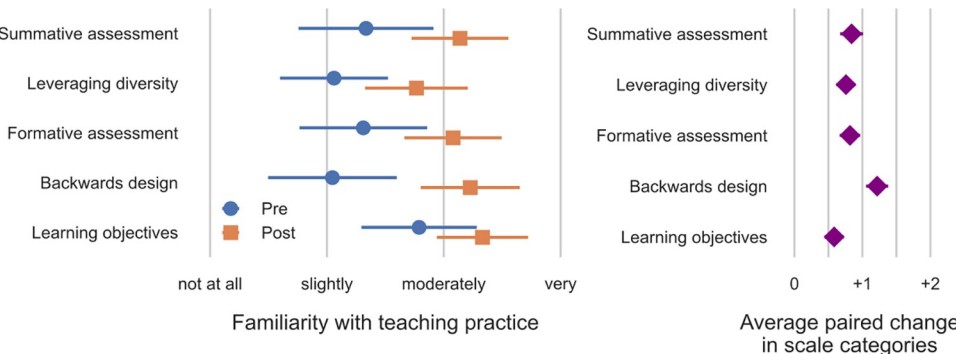

**Fig 3. Increase in average reported familiarity with pedagogical topics.** (A) Average responses of pre- and post-course respondents, unpaired. Error bars represent one standard deviation of the response distribution in each direction. (B) Average of paired differences for the 520 respondents who took both the pre- and post-course surveys for course instances where responses can be linked. Error bars represent the 99% confidence interval on the difference.

The high rate of learner completion and their high self-reported learning gains, especially among those who self-identified as future STEM faculty, indicates that the course design matched these self-directed learners' time commitment, work level, availability, and motivation. This conclusion is corroborated by satisfaction of post-course survey respondents, 97% of whom agreed that the course improved their ability to teach, 93% were either "satisfied" or "extremely satisfied" with the course, and 97% would "recommend [the course] to others"". These statistics are based on post-survey respondents to all seven instances of the introductory course.

## Learning community engagement

Finally, to answer our third Guiding Question, we explored participant and facilitator experiences in the MCLCs. Many learners engaged in our blended model of delivery: 134 institutions have hosted at least one MCLC, and many have hosted multiple times, yielding 236 total MCLCs as of Spring 2018. According to MCLC facilitator feedback, MCLCs had on average 12 participants, who were largely (75%) STEM PhD graduate students and postdocs, and who completed the course at a high rate (65%).

We do not have data on how many learners were in MCLCs., Our best estimate is that between 20–40% of our learners were in MCLCs based on data about their intentions; 33% of post-survey respondents reported participating in an MCLC, and 96% were learners in the course. The top reasons they wished to engage in local, in-person learning were the opportunities to "interact with peers" (35%), to "discuss course materials and assignments" (32%), to "meet others interested in teaching and learning" (31%), and to "receive feedback on my teaching and learning practices" (28%). Among post-course survey respondents, 34% reported participating in an MCLC, and those who did had strong outcomes: 97% were learners and 87% completed the course, representing 19% of all completers. This suggests that MCLC participation, where available, is supportive of course completion.

## Feedback from learning community facilitators

Survey responses and interviews with MCLC facilitators provided valuable feedback on the structure and efficacy of the MOOCs broadly and the MCLCs in particular. Their thoughtful feedback informed substantial revisions of the introductory and advanced MOOCs. In their MCLCs, facilitators reported using the facilitator guide and finding most Guide components

to be useful (~75%); they reviewed the guide to get ideas and used different activities to meet the needs of their particular group. Activities involving reflection, discussion or extension of course material were well received, while those that relied on participants' past teaching experience, or required peer feedback, additional reading, or reflection outside the MCLC meeting, were generally harder to implement. Both facilitators with prior expertise in the MOOC content and those without prior knowledge reported success in leading MCLCs: experts tended to prepare MCLCs as mini-courses enriched with their own content and activities, while novices conducted MCLCs in the form of peer-led study groups, largely drawing on the MOOC materials and the facilitator guide. That novice and expert leaders can lead MCLCs with the support of the Guide, makes the MCLC model sustainable and adaptable in numerous settings. Moreover, most (45 of 51) facilitator survey respondents reported they would facilitate an MCLC again, saying, for example, *"I enjoyed facilitating the MOOC, learning from it, and sharing my experience with the participants in our learning community,"* and *"It's one of my favorite things to do, even though I am doing it as a volunteer."*

## Discussion

Our purpose in this study was to explore through discovery-based research whether (1) our model enrolled participants of future STEM faculty at a scale beyond that typically reached by traditional on-campus or synchronous professional development programs; (2) to what extent did participants increase their knowledge of and confidence in delivering effective STEM instruction; and (3) examine the effects the MCLCs had on participation and participant outcomes in the broader MOOC. Our key finding in that our model of multiple offerings of two MOOCs on evidence-based undergraduate STEM teaching, intentional support for facilitated MCLCs, and open access to course materials, have successfully met needs among graduate students and postdocs for pedagogical professional development that often go unmet by traditional on-campus resources and events [45]. A number of key factors led to this result.

Our target audience of STEM graduate students and postdocs have clearly identified professional development goals and are geographically clustered at research universities. This enabled the formation of local MCLCs, since potential participants studied and worked in proximity to each other. Having local MCLCs at universities also made publicity and recruitment easier for our blended delivery mode, since the opportunity to join MCLCs could be advertised by supportive university faculty and staff members, graduate schools, departments, and centers for learning and teaching. Those faculty and staff also made ready facilitators for MCLCs. Facilitators' self-reported experience, expertise, and the similar professional goals of participants, lent MCLCs a structure and coherence that, we suggest, distinguished them from the ad hoc student meet-ups that are common in many MOOCs, leading to greater course completion rates by MCLC participants.

Participants had flexible options for engaging with course materials and resources. Some learners completed the courses by submitting quizzes and peer-graded assignments, some audited the courses by consistently watching videos, while other learners viewed course videos in an ad hoc manner on YouTube and the project website. MCLCs provided a professional development option for those who also wanted an in-person experience. For graduate students and postdocs often constrained by time, advisor priorities and the need to focus on research, these options enabled motivated future faculty to seek out and obtain pedagogical professional development on their own terms.

In addition to the stand-alone MOOC and MCLC delivery modes, course materials have been made available as open educational resources (OER) on our project website and a YouTube channel to encourage adoption and adaptation by those individuals and institutions

involved in future STEM faculty professional development. By making access as broad and simple as possible (all materials including the MCLC Facilitator Guide are freely available), we limited our ability to track use of the materials in any great detail. Statistics from our YouTube channel indicate that our 130 videos were viewed collectively 60,540 times outside the courses during the first three years of course offerings. Multiple colleagues and educators have expressed an interest in using our course materials for professional development programs at their institutions, and some have reported back on particular uses, including developing or supporting university-level teaching certificate programs, redesigning curricula, incorporating additional materials into existing educational workshops or for credit courses, and providing online professional development opportunities for diverse audiences.

Our MOOC initiative was launched from, developed by, and continues to be hosted by STEM faculty, educators, administrators, and educational developers through the CIRTL Network despite the fact that long-term sustainability is often not an outcome of NSF-funded educational initiatives [46]. The CIRTL Network brought together the initial team that developed the MOOCs; it provided a range of STEM education practitioners and researchers who contributed to the course content through interviews, resource sharing, module development, and feedback; and Network institutions hosted approximately one third of the MCLCs. While numerous individuals from outside the CIRTL Network contributed in significant ways as well, particularly as MCLC facilitators, the existing network functioned as a community of practice that enabled the initiative to succeed. From 2018 to now, the CIRTL Network has assumed all management of the MOOCs and MCLCs, which continue to be well enrolled. In this project, the CIRTL Network was instrumental, highlighting that other professional networks such as disciplinary societies, formal and informal, could serve similar design, dissemination, and sustained support functions.

Our completion rate of 11.5% is more than double the rate reported for other non-professional and non-degree MOOCs [24, 40, 41]. Recent research shows that, while general MOOC participation and completion rates have declined over the last five years, MOOCs designed for highly motivated students pursuing professional development have thrived [25]. Our findings are consistent with this trend and point to potential future uses of MOOCs and MCLCs for career and professional development needs. Asynchronous, online learning in conjunction with synchronous, in-person learning is a structure with potential to be effective in professional development domains beyond teaching, including leadership, conflict resolution, responsible conduct of research, and mentoring, as well as interdisciplinary domains such as data visualization or computational thinking.

The fact that current STEM faculty also took our MOOCs and participated in MCLCs suggests that this structure is also useful for academics especially at institutions without extensive faculty development programs. Indeed, other initiatives have leveraged our model. A recent example is the Inclusive STEM Teaching Project, a similarly structured, blended delivery course with asynchronous online content through edX and project-trained local learning community facilitators [47]. In addition, the NIH-funded Postdoc Academy project developed two asynchronous online courses targeting postdoc professional development, again also offered with local learning communities called Postdoc Academy Learning Sessions (PALS) [48, 49].

## Limitations and recommendations for future work

This study draws on data gathered for program evaluation to make claims about the utility and reach of an asynchronous online course and associated local learning communities and materials to support teaching professional development at scale, and thus it has some limitations in comparison to other literature on professional development. First, we did not attempt to

investigate variation in course activity across group demographics, career stage, or home institutional type. Early on we chose to limit collection of identifiable data, seeking to increase participation and response rates by protecting participant anonymity as much as possible. Future large scale professional development programs that wish to explore such differences may make a different choice. A second limitation, common in professional development and MOOCs, is that we rely on participants' self-assessment of learning gains and do not assess learning through exams or other external measures. Our goal was to increase the confidence of mostly future faculty in developing and applying teaching strategies, however, and confidence can only be assessed by self-report. Moreover, though there is a large overlap between learners and survey respondents, they are not the same. Our report of gains in familiarity with course topics is based on 520 people whose pre- and post-course survey responses could be paired, which is a relatively large sample, but is nonetheless a subset of course learners who may not be fully representative. Future projects should consider a longitudinal research effort to examine learning and application to classroom practices, as some have begun to do [8], and should consider multiple measures of such outcomes [50–52]. We acknowledge the substantial complexity, effort and cost of such studies.

## Conclusion

We demonstrated the effective delivery of pedagogical professional development to future STEM faculty with the potential to significantly impact undergraduate STEM education. Our design combines flexible, asynchronous content in conjunction with optional, supported and facilitated in-person learning communities, all offered within the context of a network of STEM faculty and educational developers. Our model can successfully be used and leveraged in many contexts to overcome barriers where learners seek significant professional development in constrained settings to help them meet their diverse career and professional goals.

## Supporting information

**S1 File.**
(DOCX)

## Acknowledgments

We gratefully acknowledge our many colleagues who contributed to this effort by developing MOOC modules or content to components of modules and our MOOC Learning Community facilitator's guides. Your knowledge, insights, creativity, and delivery of this material was invaluable to the success of this project and for reaching literally thousands of future faculty. We specifically would like to thank (in alphabetical order): C. Brame, S. Chasteen, M. DiPietro, C. Fata-Hartley, A. Little, J. Littrell, R. M. Mathieu, T. McMahon, and K. Spilios, for their many critical contributions.

## Author Contributions

**Conceptualization:** Bennett B. Goldberg, Derek O. Bruff, Robin McC. Greenler, Katherine Barnicle, Sandra L. Laursen, Henry (Rique) Campa, III.

**Data curation:** Katherine Barnicle, Noah H. Green, Lauren E. P. Campbell, Sandra L. Laursen, Matthew J. Ford, Claude Mack, Christina Maimone.

**Formal analysis:** Bennett B. Goldberg, Derek O. Bruff, Sandra L. Laursen, Matthew J. Ford, Christina Maimone, Henry (Rique) Campa, III.

**Funding acquisition:** Bennett B. Goldberg, Derek O. Bruff, Katherine Barnicle, Henry (Rique) Campa, III.

**Investigation:** Bennett B. Goldberg, Derek O. Bruff, Robin McC. Greenler, Noah H. Green, Sandra L. Laursen, Amy Serafini, Tamara L. Carley, Henry (Rique) Campa, III.

**Methodology:** Robin McC. Greenler, Lauren E. P. Campbell, Claude Mack, Tamara L. Carley, Henry (Rique) Campa, III.

**Project administration:** Bennett B. Goldberg, Derek O. Bruff, Katherine Barnicle, Noah H. Green, Amy Serafini, Claude Mack, Tamara L. Carley, Henry (Rique) Campa, III.

**Resources:** Lauren E. P. Campbell, Henry (Rique) Campa, III.

**Software:** Noah H. Green, Matthew J. Ford, Claude Mack.

**Supervision:** Bennett B. Goldberg, Derek O. Bruff, Henry (Rique) Campa, III.

**Visualization:** Noah H. Green, Matthew J. Ford, Christina Maimone.

**Writing – original draft:** Bennett B. Goldberg, Derek O. Bruff, Robin McC. Greenler, Sandra L. Laursen, Matthew J. Ford, Christina Maimone, Henry (Rique) Campa, III.

**Writing – review & editing:** Bennett B. Goldberg, Derek O. Bruff, Robin McC. Greenler, Katherine Barnicle, Noah H. Green, Amy Serafini, Henry (Rique) Campa, III.

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
