## [Decision Letter · Decision Letter 0]

21 Nov 2022

PONE-D-22-27145Preparing future STEM faculty nationwide through flexible teaching professional developmentPLOS ONE

Dear Dr. Goldberg,

Thank you for submitting your manuscript to PLOS ONE. After careful consideration, we feel that it has merit but does not fully meet PLOS ONE’s publication criteria as it currently stands. Therefore, we invite you to submit a revised version of the manuscript that addresses the points raised during the review process.

Please submit your revised manuscript by 5 January 2023. Please include the following items when submitting your revised manuscript:A 'Response to Reviewers' letter that responds to each point raised by the academic editor and reviewer(s). You should upload this letter as a separate file labeled 'Response to Reviewers'.A marked-up copy of your manuscript that highlights changes made to the original version. You should upload this as a separate file labeled 'Revised Manuscript with Track Changes'.An unmarked version of your revised paper without tracked changes. You should upload this as a separate file labeled 'Manuscript'.

We look forward to receiving your revised manuscript.

Kind regards,

Prof. Ritesh G. Menezes, M.B.B.S., M.D., Diplomate N.B.

Academic Editor

PLOS ONE

Journal Requirements:

Reviewers' comments:

Reviewer's Responses to Questions

**Comments to the Author**

1. Is the manuscript technically sound, and do the data support the conclusions?

Reviewer #1: Yes

Reviewer #2: Partly

2. Has the statistical analysis been performed appropriately and rigorously? 

Reviewer #1: Yes

Reviewer #2: No

3. Have the authors made all data underlying the findings in their manuscript fully available?

Reviewer #1: Yes

Reviewer #2: Yes

4. Is the manuscript presented in an intelligible fashion and written in standard English?

Reviewer #1: Yes

Reviewer #2: Yes

5. Review Comments to the Author

Reviewer #1: This is a very well written article about an impressive professional development program. I have just a few comments I hope the authors will consider in a revision:

- In the title, the term "nationwide" is not specific to which nation so that should be added as this is an international journal.

- Perhaps I missed it, but were there any "research questions" presented and answered? At times the results felt overwhelming and lacking direction as I wasn't sure what knowledge gap they were filling. Please specify what some of the questions were prior to the start of the initiative and if/how they were answered.

- I was troubled by the use of the label "learners" for the "participants who complete at least two quizzes, complete a PGA, or watch videos from at least three of the six course modules". The label learners implied that they learned something, but really they "participated", so why not call them participants? I think that label is misleading somewhat.

- In the discussion on line 278 the authors stated, "Our target audience of STEM graduate students and postdocs"... perhaps I missed it, but was this identified as the target audience earlier in the paper? Were the demographic or job characteristics of the participants in this study described? That would be helpful information.

Reviewer #2: In this study, the authors are overall trying to demonstrate the effectiveness of the model of using MCLCs to accompany a MOOC in improving and sustaining learner engagement. The participation tied to survey data seems to make this case but paper is challenging to read and follow at times, which may be remedied in part with some substantial revisions to Methods section, clarification of terms and definitions used throughout the Results, as well as a more thoughtful and focused effort to reference supplementary materials relevant to the study. There also seemed to be reliance on membership in the CIRTL network for effective implementation of MCLCs. The authors may want to consider how aspects of the model itself can be adopted more broadly in the absence of the CIRTL network.

Below are a few specific comments and recommendations for revisions:

Line 45-46: This reviewer suggests that the authors consider moving away from a deficit-based description of the equity and opportunity gaps we see in higher education with reference to student “performance”, swapping this term instead for “outcomes”. Also suggest authors avoid labeling communities of color as “minorities” as it implies status as opposed to population percentage. Suggest rephrasing this and other sentences throughout manuscript. For example, “… reduce the disparities in outcomes between marginalized students who are historically underrepresented in STEM and their [white] peers”. Or “… reduce the disparities in outcomes between STEM students in racially minoritized groups and those in the dominant group”. Please see this article published by the National Association of Hispanic Journalists for context and other ideas for alternative phrasing ideas when referring statistics and data describing communities of color.

Link: https://nahj.org/2020/08/04/nahj-asks-newsrooms-to-drop-the-use-of-minority/

Line 97: Materials and Methods. The authors did a nice job describing the structure of the MOOC and MCLCs. What’s missing, however, are the methodological and analytical details of the actual research study (some are buried in the Results but not in a consistent way). In other words, they need to provide a description of the methods by which they conducted their research on the MOOC and MCLC participants (data collection and analysis). What was the protocol for administering the pre- and post-surveys? Can the authors provide the IRB-approved survey instrument in the supplementary materials? How were the data analyzed? What statistical tests were undergone to examine the validity of their results? What was the protocol for interviewing MCLC facilitators? Were they asked specific questions? There also seemed to be some informal assessment done of OER utility from colleagues (line 266-267) as well as analytics pulled from the MOOC website itself (line 268). None of this methodological detail is provided thus making the Results, overall, difficult to follow or evaluate.

Line 110: Can the authors explain how they “encouraged” participation in the MCLC? Some additional details on how this was operationalized would be helpful.

Line 116: Curious as to why access to the MCLC Facilitator Guides requires sign-up to be a facilitator. Are there communications or training that occurs with facilitators that necessitate limiting access to this resource? On the website (https://www.stemteachingcourse.org/learning-communities/mclc-facilitators), the Q/A video for facilitators states, “Video unavailable. This video is private.”

Line 133-134: What was the distribution of participants from the 60 different countries? Were the majority in the US and Canada given the institutional composition of the CIRTL Network? Is this an important demographic to point out with respect to the findings reported in this paper?

Table 1: Mean % of learning completing the course and auditing the course do not add up to 100%. How did the authors characterize the remaining 4% in intro course and 21% in advanced course? Are these non-completers but still Learners based on definitions provided? Perhaps it would be helpful to define a non-completer in the table legend. According to Figure 1b, maybe these are the “Disengaged” and/or “Learner-other”? Or are the “Disengaged” the “non-learners (Line 171). It’s not clear. These terms need to be defined and used consistently as the subsequent sections of the paper refer to these terms when discussing survey results.

Line 150: Reference to “course completion” is not consistent with descriptions in Table 1 legend.

Line 149: The amount of information and data analysis provided in the supplementary materials is substantial. And these three lines (147-149) in the primary manuscript do not really do the material justice. If this information is critical to the findings shared in the primary manuscript, this reviewer would like to see it summarized with reference to relevant supplementary tables and figures. One might argue that the participation, activity, and outcome data could be a separate paper and subject to its own independent review. It would be helpful if the authors could focus on what information from the supplementary materials is most critical to the findings of the primary manuscript.

Line 148: In referencing citation #30, please check formatting. Is this supposed to be a footnote?

Lines 161-167: Figure 1 panel (b) – define “Disengaged”, “Learner – other”, “Learner – auditor” and “Learner-completer”. These terms are only partially consistent with Table 1 or what is referenced in the text (Lines 155-157, 171-172). This lack of consistency is making it difficult to follow the text in reference to the figures and table.

Line 213: Since authors have a figure with two panels (a and b), the text should refer to each panel separately and discuss the significance of these differences, not just the apparent differences.

Line 236: In referencing citation #35, please check formatting. Was this also intended to be a footnote?

Line 272: The Discussion provides a nice summary with some of the details that were missing in earlier parts of the paper. Suggest moving some of this information such as operationalization of MCLC recruitment (Line 110). Would like to see the author address some of the limitations of the study in this section as there were, for instance, issues with IRB approval linking demographics with survey data. This impacted the sample size and subsequent analysis.

Additional comments: There was a wordle on page 27 and several supplementary figures (S1-S9). No reference to this information was provided in the primary text; the supplementary figures are discussed in the supplementary file referenced in line 149 of the primary text. The volume of information provided in the supplement is excessive. This reviewer recommends that the authors curate this information and include only that which is relevant to the findings of the primary manuscript with specific and thoughtful reference to supplementary figures and tables in the primary text. Such efforts would help to create a more focused read of the research study and appreciation of the findings from readers.

6. PLOS authors have the option to publish the peer review history of their article (what does this mean?). If published, this will include your full peer review and any attached files.

Reviewer #1: No

Reviewer #2: No

---

## [Author Response · Author response to Decision Letter 0]

11 Jan 2023

Please see "Response to reviewers."

---

## [Decision Letter · Decision Letter 1]

12 Mar 2023

PONE-D-22-27145R1Preparing future STEM faculty through flexible teaching professional developmentPLOS ONE

Dear Dr. Goldberg,

Thank you for submitting your manuscript to PLOS ONE. After careful consideration, we feel that it has merit but does not fully meet PLOS ONE’s publication criteria as it currently stands. Therefore, we invite you to submit a revised version of the manuscript that addresses the points raised during the review process.

We look forward to receiving your revised manuscript.

Kind regards,

Prof. Ritesh G. Menezes, M.B.B.S., M.D., Diplomate N.B.

Academic Editor

PLOS ONE

Reviewers' comments:

Reviewer's Responses to Questions

**Comments to the Author**

1. If the authors have adequately addressed your comments raised in a previous round of review and you feel that this manuscript is now acceptable for publication, you may indicate that here to bypass the “Comments to the Author” section, enter your conflict of interest statement in the “Confidential to Editor” section, and submit your "Accept" recommendation.

Reviewer #3: (No Response)

2. Is the manuscript technically sound, and do the data support the conclusions?

Reviewer #3: Partly

3. Has the statistical analysis been performed appropriately and rigorously? 

Reviewer #3: Yes

4. Have the authors made all data underlying the findings in their manuscript fully available?

Reviewer #3: Yes

5. Is the manuscript presented in an intelligible fashion and written in standard English?

Reviewer #3: Yes

6. Review Comments to the Author

Reviewer #3: Thank you for submitting your revised manuscript for reconsideration. It is apparent that a lot of hard work has gone into this work. The previous reviewer had indicated that the objectives were unclear and you had added these in the introduction. However, upon reviewing your paper the objectives remain unclear and particularly how your data addresses these questions remains unclear. I re-read the abstract after reviewing your manuscript to better understand your objectives and how these were met - but was unable to find a focus even after re-reading the abstract. Consider what is the hypothesis being tested in your paper and how this was proven/disproven by your data. Additionally, the manuscript is long and may benefit from being shortened to be succinct and focus on the primary question(s) being asked/addressed.

My constructive / focused suggestions to address my above comments are as follows:

1. I would firstly recommend that the authors re-write the abstract. The abstract has too many words dedicated to background. Background is generally permitted to be 1 sentence. Methods should be briefly stated - looks like there is some attempt to do this. No results are presented in the abstract and this is the primary purpose of the abstract. Please revise the abstract and provide the pertinent results which address the objectives that you are trying to address. The abstract should end with 1-2 sentences addressing how the findings address the objectives and/or their real world applicability. The abstract should be able to stand alone and be informative to the reader providing pertinent information.

2. The methods section does clearly not state how the data will be analyzed to answer the questions posed by the authors as their objectives. This should be revised accordingly.

3. Consider revising your results so that the subsections are designed to address each of the hypotheses/objectives being tested. This may provide the reader with better clarity to see the data addressing each of the questions you are testing.

4. Depending on the above, I would suspect that the results will need to be revised around the above structure.

5. It is traditional to place the limitations section after the discussion, and to add a conclusions section to the manuscript. The conclusion section would be a short paragraph to summarize succinctly how your manuscript addressed the questions/objectives/hypotheses and the real world appliability of the data you are presenting

6. Consider removing any data and text that do not directly address the objectives/hypotheses. While I acknowledge that a lot of work has been done in this project by your team, for the reader it is very important that the manuscript clearly defines objects and whether these were met or not. Some of this additional text may be moved to the discussion where pertinent.

Minor comments:

A. Consider not using abbreviations as key words. Abbreviations are not generally referenced by academic search databases. - I would ask that the editor/editorial staff overrule this comment if abbreviations as keywords are acceptable for use as keywords for PLOS ONE

B. The authors present data with "Absolute number/percentage". This style is not denoted in the methodology. The more usual way of presenting this is "absolute number (%)". Consider revising for clarity as when reading the data it is unclear if you are reporting 2 different percentages, vs reporting a fraction, vs reporting absolute number/percentage as I believe is your intention.

7. PLOS authors have the option to publish the peer review history of their article (what does this mean?). If published, this will include your full peer review and any attached files.

Reviewer #3: No

---

## [Author Response · Author response to Decision Letter 1]

13 Apr 2023

Response to reviewer April 2023

PONE-D-22-27145

Title: Preparing future STEM faculty through flexible teaching professional development

Dear editor and reviewer,

We very much appreciate your close reading of the manuscript and we took very much to heart your questions and suggestions for revisions. As you can see by the responses below, we agreed with you, and made changes to the manuscript which we believe, and hope you will too, provide a significant improvement in clarity, connectivity, and readability. 

This response is organized in the following way: Each of the major changes (small wording or grammatical changes have not been highlighted) is listed in tabular format together with the page number from the tracked-changes resubmitted manuscript. These are in order of appearance in the manuscript. 

Change # Page # Change

1 2 As advised, the abstract was revised to remove background, add additional results, and 1-2 sentences to connect to real-world applications of the outcomes of our study. 

2 5 A sentence has been added to the guiding questions to identify for the reader the distinction between a hypothesis-driven versus an educational project with evaluation results. 

3 6 Two sentences at the start of the ‘Methods’ section clarify for the reader the following sections and their connection to each of the guiding questions. 

4 8-9 As advised, the Methods have been revised to identify which data source is associated with which guiding question and combined with the analysis section to clarify how the analysis was applied to answer the guiding questions.

5 12 As advised, the data displayed as (##/##%) has been revised and clarified. The number format refers to pre- and post-survey respondents, which has now been highlighted in the text. 

6 13 In the Results, edited the start of the section answering guiding question three to clarify the connection between results and answering the guiding question. 

7 19 As advised, limitations and implication for future work was moved to follow the discussion

8 20 As advised, one short paragraph taken from the discussion is repurposed as a conclusion. 

9 12 As advised, removed some extraneous text from the Results

Comments by reviewer with itemized changes: 

Reviewer #3: Thank you for submitting your revised manuscript for reconsideration. It is apparent that a lot of hard work has gone into this work. The previous reviewer had indicated that the objectives were unclear and you had added these in the introduction. However, upon reviewing your paper the objectives remain unclear and particularly how your data addresses these questions remains unclear. I re-read the abstract after reviewing your manuscript to better understand your objectives and how these were met - but was unable to find a focus even after re-reading the abstract. Consider what is the hypothesis being tested in your paper and how this was proven/disproven by your data. Additionally, the manuscript is long and may benefit from being shortened to be succinct and focus on the primary question(s) being asked/addressed.

The following revisions were made: 

1. Revised abstract. Change #1.

2. Explanation in the text that this is not a hypothesis-driven study but rather a program evaluation with emergent themes and conclusions based upon findings. Change #2.

3. Revise how the objectives are connected to the data. Changes #3 and #4.

4. Removed extraneous results to the supplement. Change #9. 

My constructive / focused suggestions to address my above comments are as follows:

1. I would firstly recommend that the authors re-write the abstract. The abstract has too many words dedicated to background. Background is generally permitted to be 1 sentence. Methods should be briefly stated - looks like there is some attempt to do this. No results are presented in the abstract and this is the primary purpose of the abstract. Please revise the abstract and provide the pertinent results which address the objectives that you are trying to address. The abstract should end with 1-2 sentences addressing how the findings address the objectives and/or their real world applicability. The abstract should be able to stand alone and be informative to the reader providing pertinent information.

1. Edited the abstract to cut the background and provide more results, and extend the outcomes to identify the real-world applicability. Change #1.

2. The methods section does clearly not state how the data will be analyzed to answer the questions posed by the authors as their objectives. This should be revised accordingly.

1. Edited the methods section to more clearly state this. Changes #3 and #4. 

3. Consider revising your results so that the subsections are designed to address each of the hypotheses/objectives being tested. This may provide the reader with better clarity to see the data addressing each of the questions you are testing.

1. Reorganized the results in a more straightforward way, referring directly to each of the objectives. Change #6.

4. Depending on the above, I would suspect that the results will need to be revised around the above structure.

1. See above. 

5. It is traditional to place the limitations section after the discussion, and to add a conclusions section to the manuscript. The conclusion section would be a short paragraph to summarize succinctly how your manuscript addressed the questions/objectives/hypotheses and the real world appliability of the data you are presenting

1. Moved the limitations section to after the discussion. Change #7. 

2. Edited the conclusion - repurposed the last paragraph of discussion. Change #8. 

6. Consider removing any data and text that do not directly address the objectives/hypotheses. While I acknowledge that a lot of work has been done in this project by your team, for the reader it is very important that the manuscript clearly defines objects and whether these were met or not. Some of this additional text may be moved to the discussion where pertinent.

1. Removed extraneous sections of the results. Change #9. 

Minor comments:

A. Consider not using abbreviations as key words. Abbreviations are not generally referenced by academic search databases. - I would ask that the editor/editorial staff overrule this comment if abbreviations as keywords are acceptable for use as keywords for PLOS ONE

1. Asking the editor if “STEM” and MOOC” are allowed as a keywords. 

B. The authors present data with "Absolute number/percentage". This style is not denoted in the methodology. The more usual way of presenting this is "absolute number (%)". Consider revising for clarity as when reading the data it is unclear if you are reporting 2 different percentages, vs reporting a fraction, vs reporting absolute number/percentage as I believe is your intention.

1. Revised the section that has the (number/number % sign) which actually refers to the same percentage, just one is pre- and one post-course survey. Change #5.

---

## [Decision Letter · Decision Letter 2]

26 May 2023

PONE-D-22-27145R2Preparing future STEM faculty through flexible teaching professional developmentPLOS ONE

Dear Dr. Goldberg,

Thank you for submitting your manuscript to PLOS ONE. After careful consideration, we feel that it has merit but does not fully meet PLOS ONE’s publication criteria as it currently stands. Therefore, we invite you to submit a revised version of the manuscript that addresses the points raised during the review process.

We look forward to receiving your revised manuscript.

Kind regards,

Prof. Ritesh G. Menezes, M.B.B.S., M.D., Diplomate N.B.

Academic Editor

PLOS ONE

Reviewers' comments:

Reviewer's Responses to Questions

**Comments to the Author**

1. If the authors have adequately addressed your comments raised in a previous round of review and you feel that this manuscript is now acceptable for publication, you may indicate that here to bypass the “Comments to the Author” section, enter your conflict of interest statement in the “Confidential to Editor” section, and submit your "Accept" recommendation.

Reviewer #1: All comments have been addressed

Reviewer #3: All comments have been addressed

2. Is the manuscript technically sound, and do the data support the conclusions?

Reviewer #1: Yes

Reviewer #3: Partly

3. Has the statistical analysis been performed appropriately and rigorously? 

Reviewer #1: Yes

Reviewer #3: N/A

4. Have the authors made all data underlying the findings in their manuscript fully available?

Reviewer #1: Yes

Reviewer #3: Yes

5. Is the manuscript presented in an intelligible fashion and written in standard English?

Reviewer #1: Yes

Reviewer #3: No

6. Review Comments to the Author

Reviewer #1: Excellent work on revisions, this manuscript is ready for publication. I verify that all required questions have been answered and that all responses meet formatting specifications.

Reviewer #3: Thank you for your revisions. These have improved the readability of your paper. Unfortunately, despite changes your manuscript remains somewhat challenging to follow. I would suspect that your decision to pursue "guiding questions" rather than "formal hypotheses" significantly impairs the readability of the paper. This is because your key message(s) are not clearly stated making it particularly hard to define the purpose of the paper and key messages. To be useful to the reader, the authors must clearly state their purposes and ensure that the key take away points are clearly stated for the reader at the start of the discussion and at the conclusion of reading the paper.

Some key points to address:

Abbreviations STEM and MOOC should not be used in keywords - this has not been addressed.

Line 205 : The average introductory course completion rate of 11.5% overall is more than double the rate reported for other non-professional and non-degree MOOCs [34, 35, 25]. Belongs in discussion as this is not your results but rather a discussion of your results

Line 207: You report an "advanced MOOC" but this is never / not clearly defined in your methodology. Please specify in the methods that there are 2 MOOCs being evaluated and how these differ.

Line 215: "substantially outranked achievement oriented goals including “earn credentials for my CV” and “earn a statement of accomplishment.”" but the percentaged for both these are not specified. They should be specified in the results to support your statement that these were "substantially" different

Line 233 The "threshold" for "drop-offs" is undefined in the methodology

Line 252-3 : "Pre- and post-course respondents were largely PhDs and postdocs (50% pre-course and 58% post course) with faculty an additional 2%." Who were the other 48%?

Line 284: "Willingness" should be lower case

Line 340-352: Unclear why this "Open educational resource engagement" section is in the results section. Not sure what results are being shared here. Does this belong in the discussion?

Discussion: In your opening paragraph you need to sumarize what your key findings were. Based on your format of using guiding questions, this should be a 3-4 line paragraph which summarizes clearly what your take key results/findings of your study are. Tyipcally this would be a place where you would discuss your hypotheses and if they were proven or disproven.

Abbreviations:

- postdocs - should be expanded

- MOOCs - needs to be defined in body of manuscript not just abstract

7. PLOS authors have the option to publish the peer review history of their article (what does this mean?). If published, this will include your full peer review and any attached files.

Reviewer #1: No

Reviewer #3: No

---

## [Author Response · Author response to Decision Letter 2]

5 Jun 2023

Response to reviewers June 2023

PONE-D-22-27145

Title: Preparing future STEM faculty through flexible teaching professional development

Dear editor and reviewers,

We are pleased that Reviewer #1 said, “Excellent work on revisions, this manuscript is ready for publication.” We appreciate the comment and the confirmation that Reviewer #1 feels we met the expectations with the prior revisions.

We are pleased as well that Reviewer #3 saw significant improvement. We very much appreciate the close reading of the manuscript and we took very much to heart the remaining suggestions for improvement and publication. As you can see below, we made changes to the manuscript based on reviewer comments which we believe, and hope you will too, provide a significant improvement in clarity, connectivity, and readability. 

This response is organized in the following way: Each of the major changes is listed in tabular format together with the page number from the tracked-changes resubmitted manuscript. These are in order of comment by Reviewer 3 and close to their order of appearance in the manuscript. 

Change # Page # Change

0 5 & 6 We significantly revised the end of the introduction to specifically describe the purpose of our work, and to contextualize that purpose within a wider body of literature. Similarly, we clarify the distinction between hypothesis-driven research and real-world, discovery-based research.

1 3 STEM and MOOCs have been spelled out in the key words

2 3 & 18 The number has stayed on p. 9, but the comparison with prior research and courses has moved to the discussion on p. 18. 

3 6 The opening line of methods has been modified to specifically indicate the the two courses, an introductory and advanced MOOC will be described in methods under “Course structure and logistics”

4 10 These percentages have been added to p. 10. In addition, the manuscript has been corrected to refer to S6 Table, not S6 Fig. 

5 10 The threshold is now defined on p. 11, an oversight on our part.

6 12 On p. 12 the faculty number was corrected to 20%, not 2%. In addition, the other category percentages and a description were added. 

7 13 “willingness” has been changed to lower case on p. 13, apologies for the typo.

8 17 The section on "Open educational resource engagement" has been moved to the discussion on p. 17

9 71 As recommended, an opening paragraph has been added on p. 17 to the discussion that reiterates the context of this work as discovery-based research, explicates the guiding questions and specifies the key findings. 

10 3 & 4 The first instance of ‘postdocs’ on p. 3 is expanded, and explicated as ‘postdocs’ hereafter. The first instance of MOOCs on p. 4 in the manuscript had already been expanded and attached to the commonplace acronym. 

Comments by reviewer with itemized changes: 

Reviewer #1: 

Reviewer #1: Excellent work on revisions, this manuscript is ready for publication. I verify that all required questions have been answered and that all responses meet formatting specifications.

Thank you! 

Reviewer #3: 

“Thank you for your revisions. These have improved the readability of your paper. Unfortunately, despite changes your manuscript remains somewhat challenging to follow. I would suspect that your decision to pursue "guiding questions" rather than "formal hypotheses" significantly impairs the readability of the paper. This is because your key message(s) are not clearly stated making it particularly hard to define the purpose of the paper and key messages. To be useful to the reader, the authors must clearly state their purposes and ensure that the key take away points are clearly stated for the reader at the start of the discussion and at the conclusion of reading the paper.”

Change #0: We significantly revised the end of the introduction to specifically describe the purpose of our work, and to contextualize that purpose within a wider body of literature. Similarly, we clarify the distinction between hypothesis-driven research and real-world, discovery-based research that we intentionally pursue. 

“Abbreviations STEM and MOOC should not be used in keywords - this has not been addressed.”

Change #1: STEM and MOOCs have been spelled out in the key words

“Line 205 : The average introductory course completion rate of 11.5% overall is more than double the rate reported for other non-professional and non-degree MOOCs [34, 35, 25]. Belongs in discussion as this is not your results but rather a discussion of your results”

Change #2: The number has stayed on p. 9, but the comparison with prior research and courses has moved to the discussion on p. 18. 

“Line 207: You report an "advanced MOOC" but this is never / not clearly defined in your methodology. Please specify in the methods that there are 2 MOOCs being evaluated and how these differ.”

Change #3: The opening line of methods has been modified to specifically indicate the two courses, an “introductory” and “advanced” MOOC will be described in methods under “Course structure and logistics”

Additional response to reviewer #3: The methods section on p. 6 and 7 describe the detailed differences between the introductory and advanced MOOCs. 

“Line 215: "substantially outranked achievement oriented goals including “earn credentials for my CV” and “earn a statement of accomplishment.”" but the percentaged [sic] for both these are not specified. They should be specified in the results to support your statement that these were "substantially" different”

Change #4: These percentages have been added to p. 10. In addition, the manuscript has been corrected to refer to S6 Table, not S6 Fig. 

“Line 233 The "threshold" for "drop-offs" is undefined in the methodology”

Change #5: The threshold is now defined on p. 11, an oversight on our part.

“Line 252-3 : "Pre- and post-course respondents were largely PhDs and postdocs (50% pre-course and 58% post course) with faculty an additional 2%." Who were the other 48%?”

Change #6: On p. 12 the faculty number was corrected to 20%, not 2%. In addition, the other category percentages and a description were added. 

“Line 284: "Willingness" should be lower case”

Change #7: “willingness” has been changed to lower case on p. 13

“Line 340-352: Unclear why this "Open educational resource engagement" section is in the results section. Not sure what results are being shared here. Does this belong in the discussion?”

Change #8: The section on "Open educational resource engagement" has been moved to the discussion on p. 17 per the recommendation.

“Discussion: In your opening paragraph you need to sumarize [sic] what your key findings were. Based on your format of using guiding questions, this should be a 3-4 line paragraph which summarizes clearly what your take key [sic] results/findings of your study are. Tyipcally [sic] this would be a place where you would discuss your hypotheses and if they were proven or disproven.”

Change #9: As recommended, an opening paragraph has now been added on p. 17 to the discussion that reiterates the context of this work as discovery-based research, explicates the guiding questions and specifies the key findings. 

Abbreviations:

- postdocs - should be expanded

- MOOCs - needs to be defined in body of manuscript not just abstract

Change #10: The first instance of ‘postdocs’ on p. 3 is expanded, and explicated as ‘postdocs’ hereafter. The first instance of MOOCs on p. 4 in the manuscript had already been expanded and attached to the commonplace acronym. 

We hope these revisions now meet with the expectations of reviewer #3, the Associate Editor, and Editor.

---

## [Editor Report · Decision Letter 3]

6 Jun 2023

Preparing future STEM faculty through flexible teaching professional development

PONE-D-22-27145R3

Dear Dr. Goldberg,

We’re pleased to inform you that your manuscript has been judged scientifically suitable for publication and will be formally accepted for publication once it meets all outstanding technical requirements.

Kind regards,

Prof. Ritesh G. Menezes, M.B.B.S., M.D., Diplomate N.B.

Academic Editor

PLOS ONE

---

## [Editor Report · Acceptance letter]

22 Jun 2023

PONE-D-22-27145R3 

Preparing future STEM faculty through flexible teaching professional development 

Dear Dr. Goldberg:

I'm pleased to inform you that your manuscript has been deemed suitable for publication in PLOS ONE. Congratulations! Your manuscript is now with our production department. 

Kind regards, 

on behalf of

Professor Ritesh G. Menezes 

Academic Editor

PLOS ONE